# Isolation of Lineage Specific Nuclei Based on Distinct Endoreduplication Levels and Tissue-Specific Markers to Study Chromatin Accessibility Landscapes

**DOI:** 10.3390/plants9111478

**Published:** 2020-11-03

**Authors:** Ezgi Süheyla Karaaslan, Natalie Faiß, Chang Liu, Kenneth Wayne Berendzen

**Affiliations:** 1Center for Plant Molecular Biology (ZMBP), University of Tübingen, Auf der Morgenstelle 32, 72076 Tübingen, Germany; ezgi.dogan@zmbp.uni-tuebingen.de (E.S.K.); natalie.faiss@zmbp.uni-tuebingen.de (N.F.); 2Institute of Biology, University of Hohenheim, Garbenstrasse 30, 70599 Stuttgart, Germany

**Keywords:** FANS, endopolyploidy, skotomorphogenesis, photomorphogenesis, ATAC-seq, hypocotyl, hromatin accessibility

## Abstract

The capacity for achieving immense specificity and resolution in science increases day to day. Fluorescence-activated nuclear sorting (FANS) offers this great precision, enabling one to count and separate distinct types of nuclei from specific cells of heterogeneous mixtures. We developed a workflow to collect nuclei from *Arabidopsis thaliana* by FANS according to cell lineage and endopolyploidy level with high efficiency. We sorted GFP-labeled nuclei with different ploidy levels from the epidermal tissue layer of three-day, dark-grown hypocotyls followed by a shift to light for one day and compared them to plants left in the dark. We then accessed early chromatin accessibility patterns associated with skotomorphogenesis and photomorphogenesis by the assay for transposase-accessible chromatin using sequencing (ATAC-seq) within primarily stomatal 2C and fully endoreduplicated 16C nuclei. Our quantitative analysis shows that dark- and light-treated samples in 2C nuclei do not exhibit any different chromatin accessibility landscapes, whereas changes in 16C can be linked to transcriptional changes involved in light response.

## 1. Introduction

In plant development, light acts as a critical environmental modulator at nearly every stage. In order to reach the surface, seeds germinating under the soil direct resources to elongate the hypocotyl by a developmental program called skotomorphogenesis. Skotomorphogenesis is crucial for plant survival and is characterized not only by rapid hypocotyl growth, but also by a closed apical hook to protect the apical meristem from mechanical damage. On the contrary, seedlings exposed to light undergo photomorphogenesis, an antagonistic developmental progress that is characterized by a short hypocotyl and open cotyledons [1]. Such a delicate developmental balance is tightly regulated at the transcriptional level by various transcription factors cascades [2]. In *Arabidopsis*, PHYTOCHROME INTERACTING FACTORS (PIFs) are well-characterized central regulators of skotomorphogenesis. PIFs are basic helix-loop-helix transcription factors (TFs) which constitutively suppress photomorphogenesis in young seedlings by directly binding to the G-Box elements in the promoters of several target genes [3,4,5,6,7]. PIF1, PIF3, PIF4 and PIF5 play overlapping roles in promoting dark state development: in the quadruple mutant (*pifq*), skotomorphogenesis is interrupted and results in a shorter hypocotyl and open cotyledons [8,9,10]. Upon initial exposure to light, PIFs are rapidly phosphorylated by photo-activated phytochromes and subsequently degraded by the 26S proteasome, initiating photomorphogenesis [4,11,12,13]. Another transcription factor, ELONGATED HYPOCOTYL 5 (HY5), has a major role in regulating photomorphogenesis [14]. This bZIP TF interacts with light response genes by preferentially binding to G-Box motifs, but also LREs Z-CA- and CG-hybrid boxes elements at their promoter regions [3,15]. Several immunoprecipitation and transcriptome analyses have shown that HY5 regulates thousands of genes in the *Arabidopsis* genome [15,16,17]. There is emerging evidence that not only transcription factors, but also histone modifiers, are involved in the regulation of dark-to-light transition of plants. Several studies have shown opposing regulatory activities of histone acetyltransferases (HATs) and deacetylases (HDACs) following light perception [18,19,20]. It is crucial to investigate the differences in chromatin accessibility patterns throughout the hypocotyl as chromatin states and histone modifications at various plant developmental stages in response to environmental cues are gaining importance [21,22].

Endoreduplication, also called somatic polyploidization or endopolyploidy, is coupled to development in many plant species [23]. Endoreduplication occurs in all cell types except gametes, guard cells, meristematic cells and floral organs [24]. Increased DNA amount as a result of endoreduplication events is closely associated with cell growth in many organisms [25,26,27] and in dicotyledonous plants, endoreduplication in hypocotyl tissue is an essential part of skotomorphogenesis. The *Arabidopsis* hypocotyl is composed of only 20 cell files; several endoreduplication cycles in the hypocotyl tissue provide cell expansion and therefore tissue expansion [2,28]. It is known that one extra endoreduplication cycle is observed in dark-grown hypocotyls compared to light-grown hypocotyls [28], and this is inhibited via various photoreceptors during photomorphogenesis [2,28]. The phytohormones gibberellin, auxin and brassinolide clearly promote hypocotyl elongation in dark-grown plants [2,4,29], whereas abscisic acid inhibits hypocotyl elongation when gibberellic acid and auxin concentrations are low [30]. Although the exact mechanism of how ploidy levels regulate cell growth remains elusive [31], it has been proposed that increased transcriptional and metabolic activity due to the multiplication of genetic material contributes to cell growth [32,33,34,35,36,37].

Gene expression is closely connected to chromatin structure. Currently, it is unclear to what extent chromatin accessibility of individual genes varies during the developmental transition from skotomorphogenesis to photomorphogenesis is coupled with changes in endoreduplication activities. Flow cytometry can be used to purify nuclei from tissue homogenates [38], making it possible to harvest cell-type-specific nuclei from plant tissues using nuclear-targeted fluorescent proteins, as demonstrated for studying endoreduplication within specific cell types in the root [39] or exploring the transcriptome of genes selectively expressed in the phloem [40]. Studying the changes in chromatin accessibility that occur upon light exposure may provide us with a better view on how light initiates different regulatory mechanisms that are dependent on chromatin structure changes. In this study, we performed the assay for transposase-accessible chromatin using sequencing (ATAC-seq), a widely used high-throughput method to detect open chromatin regions, to assess chromatin accessibility in a tissue-, lineage- and endopolyploidy-specific manner. Such an assay on this particular tissue and cell type has never been done before.

## 2. Results

### 2.1. Isolation of Lineage-Specific Nuclei Based on Endopolyploidy Levels and a Tissue-Specific Marker for ATAC-seq

Since the association between skotomorphogenesis and cellular endoreduplication has been demonstrated in epidermal cell layers of the hypocotyl [28,41], we used an epidermal tissue-specific marker *pAtML1::NLS:GFP* line to obtain GFP-tagged epidermal nuclei from plants growing under different light conditions. Along with specifically harvesting GFP-positive nuclei by fluorescence activated nuclei sorting (FANS) [42], we further separated them according to their endopolyploidy levels and selected 2C and 16C to perform ATAC-seq (Figure 1). Three-day-old *Arabidopsis* seedlings were grown in complete darkness, then transferred into the light for one day in order to collect nuclei for light-treated samples. Seedlings were kept in the dark for an additional day to collect nuclei from dark-treated samples (Figure 1A). This 1 day exposure window allows plants to transition from skotomorphogenesis to photomorphogenesis exactly when the hypocotyl grows the most [28] (Figure 1B). Several morphological differences were observed after one day of change: upon light exposure, the cotyledons opened and hypocotyls ceased their rapid elongation, whereas the dark-grown plants practically doubled in hypocotyl length compared to the light-grown plants (Figure 1B). This is consistent with what has been previously observed [28].

Chemical fixation preserves biological materials for downstream processes [43,44]. To get the best ATAC-seq results, we fixed the hypocotyls prior to FANS using 4% formaldehyde (FA), as we found that this concentration had superior nuclear GFP signals compared to 1%, yielding clearly discernible ATAC-seq peaks useful for downstream analysis (Appendix A). After fixation, the hypocotyl was carefully excised by removing the cotyledons and roots and stored at −8 °C. To extract nuclei, the tissue was chopped with a razor blade in buffer for minimal structural disruption and filtered twice through 40 μm nylon mesh. Nuclei were stained with DAPI and analyzed by flow cytometry before FANS (Figure 1C,D). In both skotomorphogenic and photomorphogenic plants, multiple nuclear DAPI (Figure 1D–G) and GFP (Figure 1E–G) populations were observed due to endoreduplication events occurring within the hypocotyl tissue. The most extreme ploidy levels were sorted, 2C and 16C, based on their DAPI and GFP signal (Figure 1E,F). After obtaining an adequate number of nuclei by FANS, we proceeded with the Tn5 integration step of the ATAC-seq protocol.

Most of the GFP expressing nuclei were at the 8C and 16C stage (Figure 1F,G). According to the counts extracted from the FANS sort data, our analysis shows that epidermal layer cells are the main contributors of high endopolyploidy nuclei in the hypocotyl comprising ~80% and ~66% of all 16C nuclei in the dark- and light-treated samples, respectively (Figure 1H). This corroborates early microscopy-based observations that the epidermal and cortical cells contain the highest DNA levels [41]. The shift from dark to light also correlated with a reduction in the percentage of 8C and 16C endoreduplicated nuclei (Figure 1H), indicating that the switch from skoto- to photomorphogenesis leads to a halt in the endoreduplication process. These results indicate that we have collected material for ATAC-seq from the major hypocotyl cell layer involving endoreduplicated nuclei.

### 2.2. ATAC-seq Revealed Accessible Chromatin Patterns in Dark and Light Grown Hypocotyls

After performing ATAC-seq for light- and dark-treated hypocotyls in 2C and 16C nuclei, we identified roughly ~38,000 peaks across all samples. Our initial analysis of the ATAC-seq data revealed that the global chromatin accessibility landscape throughout the *Arabidopsis* genome is very similar among the different samples (Figure 2A). As several transcription factors are involved in the skoto- to photomorphogenesis transition, we analyzed local changes in chromatin accessibility upon light exposure around TF binding sites. Considering the important regulatory roles of the PIF transcription factor family and HY5 during this transition, we used previously published ChIP-seq data to analyze potential differences in chromatin accessibility around PIFs and HY5 binding sites. We selected PIF ChIP-seq data from Pfeiffer et al. [45] and HY5 ChIP-seq data from Burko and colleagues [46]. Our ATAC-seq analysis revealed no changes in local chromatin accessibility around PIF1, PIF3, PIF4, PIF5 and HY5-binding DNA regions in neither 2C nor 16C nuclei (Figure 2 and Appendix A). As not only PIF targets, but also *PIF* genes are regulated at the transcriptional level [5,47], we inspected *PIF1* and *PIF3* for differences in chromatin accessibility around their gene bodies and promoters. Like the global pattern, both *PIF1* and *PIF3* displayed similar chromatin accessibility in dark and light hypocotyl in both 2C and 16C nuclei (Appendix A).

Thereafter, we performed quantitative analysis to identify differentially accessible chromatin regions from light and dark samples in 2C and 16C nuclei (see the “Materials and Methods” section). When we compared light- and dark-treated 2C nuclei, our quantitative analysis revealed that only one region was more accessible in light-treated samples and no region was more accessible in dark-treated samples. This means light- and dark-treated 2C nuclei have identical open chromatin landscapes. Dark- and light-treated 16C nuclei, on the other hand, had a significant number of differential ATAC-seq peaks. Comparing light- and dark-treated 16C nuclei revealed, on one hand, that 70 regions that were more accessible in light samples; we will refer to them from now on as light accessible regions (LARs). On the other hand, 20 regions were more accessible in 16C dark samples, which we refer to from now on as dark accessible regions (DARs). We then determined which genes are overlapping with LARs and DARs. We considered a gene as a ‘differentially accessible gene’ if the genomic region from 1000 bp upstream of the annotated transcription start site (TSS) to its transcription termination site (TTS) is overlapping with one or more LAR or DAR. Only one gene, *AT3G44970*, encoding a cytochrome P450 superfamily protein, was more accessible in 2C light samples (Appendix A). This gene is expressed specifically in the hypocotyl according to a Kelpikova Atlas eFP Browser analysis (Appendix A). Upon light exposure in 16C nuclei, 56 genes from 54 out of 70 LARs were more accessible in the light-treated samples and 11 genes from 11 out of 20 DARs were more accessible in dark-treated samples (Appendix A). Therefore, we focused only on 16C nuclei for further analysis of differentially accessible regions. Examples of two genes with LARs are shown in Figure 3A,B.

We next performed Gene Ontology (GO) analysis on the two groups of 16C genes. The 11 genes that are more accessible in dark-treated samples did not show any GO enrichment (*p*-value cutoff set as 0.05). In contrast, GO analysis for the 56 genes more accessible upon light exposure revealed that the set is enriched for photosynthesis, response to light and metabolite processing (Figure 3C; Appendix A). The LAR genes appear to be quite diverse. Nevertheless, 14 of the 56 are core photosynthesis genes, there are several transcription factors, two genes involved in ABA signaling, and a few genes that can be linked to chromatin regulation. Five genes are also known to produce mobile signals at the RNA level [48]. The genes harboring DARs are also fairly diverse, yet three also produce growth factors or mobile signals (Appendix A).

Furthermore, in order to determine candidate transcription factors located in differentially accessible regions, we performed Multiple Em for Motif Elicitation (MEME) motif analysis [49] on LAR and DARs. Only two significant motifs were returned for LARs and none for DARs (Appendix A). The strongest hit (*p* = 2.6 × 10^−13^) was the motif “ACGTG” (Figure 3D), which is strongly bound by HY5 [46] and indirectly by PIFs [50]. The tetra-nucleotide “ACGT” is the typical core recognition sequence for plant bZIP transcription factors [51,52,53] and the “ACGTGK” motif, also called ABA-RESPONSIVE ELEMENT (ABRE), is found in promoters of many abscisic acid (ABA)-inducible genes, conferring ABA responsiveness [52,54].

### 2.3. Analysis of Identified Differentially Accessible Genes for Regulation by Light by Mining Publicly Available Expression Data

We decided to investigate if there is any evidence that the gene sets we obtained are regulated by light at the transcriptional level. For 46/56 of the LAR genes present in the AtGenExpress Light Treatments dataset, 17 of 46 of them are actually up-regulated by red, far-red, blue and/or UV-light in wild-type Col-0 seedlings. Three of the 9/11 DAR genes on the Affymetrix chip are down-regulated by different light qualities (Figure 4; Appendix A). Not a single gene was found to be two-fold or more inversely regulated in either dataset. Thus, we see that the ATAC-seq dataset faithfully captured light-regulated regions whose chromatin environment can also be mirrored in transcriptional changes.

In addition, we also compared the LAR and DAR genes with a list of genes specifically identified as light-regulated in the hypocotyl and cotyledons in *Arabidopsis* (Col-0) seedlings from Sun et al. [55]. In this RNA-seq analysis, plants were kept in continuous darkness for four days and then transferred to white light for 1 or 6 h, whereafter cotyledons and hypocotyls were dissected. Approximately 67% (38/56) of all the LARs and 63% (7/11) of the DARs are found in their complete dataset. Of the 67% LAR genes that overlap, only 7/38 are labeled as cotyledon-specific, meaning that 55% of our LAR genes are regulated specifically in the hypocotyl according to Sun et al. Similarly, applying the same analysis, we find 4/11 (36%) of the DAR genes specifically regulated in the hypocotyl (Appendix A).

If we consider only the LAR genes and compare them only to those genes regulated after 6 h of light treatment from hypocotyl tissue from the Sun et al. experiment, around 50% (28/56) of the more accessible LAR genes are actually upregulated in the RNA-seq dataset, whereas 36% (4/11) of DAR genes are down-regulated (Appendix A). Thus, the changes occurring in chromatin accessibility appear to result in transcriptional reprogramming, as the genes that are falling into LARs or DARs are also up- or down-regulated in the RNA-seq as well. This result is consistent with the AtGenExpress light experiment analysis shown above (Figure 4). We also tested the chromatin accessibility of the total list of 3400 upregulated genes in this particular RNA-seq dataset in our ATAC-seq dataset. These genes, however, displayed similar levels of chromatin accessibility in all samples, which is quite open in the region at, and directly upstream of, the TSS (Appendix A). In summary, we observe very few differential chromatin changes in the 2C and 16C of epidermal cells. Nevertheless, the genes we identified as having light-dependent chromatin accessibility changes in the 16C are paralleled in light-dependent transcriptional changes, meaning that transcriptional reprogramming can directly be associated with changes in the chromatin accessibility for at least the 16C cell lineage.

## 3. Discussion

Light is one of the most important abiotic factors regulating plant growth and development. In this study, we have established a nuclear ATAC-seq workflow that allows one to work on specific cell pools from individual tissue layers. In particular, FANS enabled us to distinguish, sort and analyze 2C and 16C endoreduplicated epidermal cells from the hypocotyl. In our analysis, we identified ~38,000 peaks by ATAC-seq, which could be mapped to 90 differentially accessible dark-to-light regions, corresponding to 67 differentially accessible genes among 16C nuclei and one in 2C nuclei. Overall, our ATAC-seq analysis reveals that global chromatin accessibility patterns among dark and light treated epidermal hypocotyl 2C and 16C nuclei did not show dramatic changes after one day of white light exposure.

It has been demonstrated that light-responsive cis-elements are localized in close proximity to the TSS [16]. PIFs and HY5 regulate ~20% of the genome, controlling hormonal pathways and themselves being controlled by phytochromes to modulate photomorphogenesis [2]. PIFs accumulate in darkness, are essential for skotomorphogenesis, and are the major suppressors of photomorphogenesis, promoting hypocotyl elongation and inhibiting chloroplast differentiation (reviewed in [56]), whereas HY5 in the light is the major repressor of skotomorphic PIF function [5]. PIFs bind to G-boxes, but also E-, PBE-boxes and G-box coupling elements [3] and similarly, HY5 binds to G-boxes, but also LREs Z-CA- and CG-hybrid boxes elements [3,51]. The HY5 binding motif stood out as a main TF binding motif in LARs (Figure 3D and Appendix A), but could be a target of both TFs. Loss of HY5 leads to a long hypocotyl under red, far-red, blue and white light [57]. Many of our 67 genes are regulated by one or more of these light qualities (Figure 4), suggestive of a direction connection to HY5. The physiological response to light in the hypocotyl was abrupt and evident after 24 h (Figure 1B); globally, however, the chromatin around TSS regions was already open such that the fate for both cell types appears to be predetermined. Using ChIP-seq against native HY5, Zhang et al. 2011 [15] reported that while it bound to ~11,000 genes under their experimental conditions, only a tenth of those had altered gene expression, and 60% of the HY5 bound regions were bound irrespective of the light/dark conditions. This could also be the case here, where the chromatin environment is already set but waiting for particular sets of controlling factors to initiate developmental programs. In this regard, it is not so surprising that we uncover only a few changes in very specific cell types then. It has been shown, however, that light signaling triggers chromatin rearrangements in different developmental stages [21]. This indicates that most likely, major changes in chromatin accessibility, and likely transcriptional regulation, occur either at the other ploidy levels in the epidermis, and/or in other cell types from the hypocotyl. Alternatively, maybe longer-term changes are recorded in the chromatin, but are not part of the early photomorphogenic response, at least in these particular cell types. Nevertheless, for those differential accessibility regions detected in 16C nuclei, we clearly detected genes that respond to light to mediate photomorphogenesis and respond to light at the chromatin level.

With regard to 2C nuclei, almost no other cell divisions in the hypocotyl occur after seed imbibition [28,29], such that the very few to no cell divisions that do occur, are only in the cortex or epidermal hypocotyl cells which are primarily 2C stomata cells [28,29,41,58]. Other 2C cells can eventually divide much later during secondary growth although the majority of cells in the central cylinder stay at 2C/4C [28,41], observed here in the non-GFP populations (see Figure 1E,G). This means most of the 2C cells we sorted from the hypocotyl epidermis were primarily already fated to be stomatal cells. We can conclude that if there are transcriptional changes due to light conditions (which we did not test), we have to predict at the moment that they must be independent of large changes in the chromatin landscape, since the 24 h light treatment did not have a strong effect on the open chromatin pattern in this cell type.

Our experimental setup was designed to capture the largest amount of exponential growth in the hypocotyl, which occurs from day three to day four in both light- and dark-grown seedlings [28]. Hypocotyl growth is overwhelmingly accomplished by cell elongation via promoting endoreduplication in the dark and repressing it in the light, mediated by various light receptors [28,59]. Thus, part of the strong response to light should include a halting of endoreduplication, and in fact we observed this in our data as well. Cells that enter the endocycle program cannot resume mitotic divisions [60]. Cell growth as such is independent of cell divisions or cell wall synthesis and therefore requires expansins [2]. Two genes are likely involved in cell wall synthesis/signaling: *ARABINOGALACTAN PROTEIN 12* (*AT3G13520)* and *L-GALL RESPONSIVE GENE 1* (*AT1G80240*) have DARs (Appendix A), suggestive of a halt to cell expansion, as we believe that DAR genes are likely downregulated in the light. While most of the genes we captured are not classically involved in endoreduplication [35], two other genes, *AT5G15960* and *AT5G45870,* are involved in ABA signaling and have LARs. This is consistent with ABA as a negative regulator of endoreduplication [2,30], since we presume that both genes were upregulated in our experiment.

Endoreduplication leads not only to larger cells, but also larger nuclei and more chromatin [61], and more complex nuclear morphology [37]. Constrained chromatin mobility increases with higher endopolyploid levels [62]. Our 16C nuclei adopted a more circular shape after one day of light exposure (Appendix A), implying some changes in genome packing with a halt to endoreduplication. LARs genes *FANCONI ANEMIA COMPLEMENTATION GROUP M* (*AT1G35530*), a highly conserved helicase and *AT4G17240,* encoding a structural maintenance of chromosomes protein might be part of this process. Two other genes, *HOMOLOG OF YEAST ADA2 2B* (*AT4G16420*), which can stimulate acetyltransferase GCN5 on free histones or nucleosomes [63] and *ARABIDOPSIS TÓXICOS EN LEVADURA 18* (*AT4G17245*), encoding a RING/U-box protein, might also take part. How these genes would lead to changes in the overall nuclear morphology is unclear at this point. Signaling through the hypocotyl between the shoot and root apical meristems occurs through the trafficking of mobile signals, which may be RNA or protein products [64], and five genes with LARs and three genes with DARs encode RNAs that cell-to-cell mobile [48] (Appendix A). However, the four LAR genes that encode components of the photosynthesis machinery are unlikely to be directly connected to endoreduplication.

We speculate that we have likely captured genes involved in halting the endocycle process, as it would have proceeded to 32C if left in the dark [65]. If we presume that the already differentiated cells cannot de-differentiate as has been suggested before [28,41], then it is likely those cells at the 8C level, or those just entering the 16C endocycle, were halted as they took on their new photomorphogenic fate. It is remarkable, then, that our experiment is able to capture such subtle changes in such a small cell pool. However, in order to get a more detailed overview about whether transcription activation triggers chromatin opening or histone remodeling attracts transcriptional activators in the hypocotyl, we need to perform further time course ATAC-seq and RNA-seq analyses, possibly also looking at the 4C and 8C nuclei. Using cortex or other cell-specific reporter lines would also be very interesting to explore, since it is conceivable that light-regulated transcriptional changes could also be occurring there. Deciphering such genome-wide information regarding photomorphogenesis and skotomorphogenesis will enhance our understanding of how the balance between these two antagonistic developmental processes is mediated to and through the chromatin environment.

## 4. Materials and Methods

### 4.1. Plant Materials and Nuclei Sorting

*Arabidopsis* used in this study were grown at 23 °C in long days (16 h light/8 h dark) on half-strength Murashige and Skoog (MS) medium [66] supplemented with 1% sucrose and 0.3% phytagel. The transgenic line used in this study was *pAtML1:NLS:3xGFP* in the Columbia-0 background [67]. For each replicate, 500 seeds were sterilized by washing with 70% ethanol for 20 min followed by extensive rinsing with water. For dark-treated samples, plants were grown on plates under the dark for 4 days. For light-treated samples, plants were grown in the dark for three days and transferred into the light for 1 day. Nuclei were isolated from fixed tissue and by flow cytometry as in [68] using a MoFlo (Beckman Coulter) with the following modifications: after initially gating nuclei from debris, each DAPI channel was compared by peak vs. integral to remove clumps and doublets, followed by inter-channel comparisons and monitoring the pulse-width of all channels to remove further clumps and debris. GFP was detected in the FL1 (530/34) via 488nm excitation. Pinhole spillover was accounted for by monitoring nuclei without GFP and sort regions adjusted appropriately. The extracted nuclei were stained with 0.5 μM DAPI to reveal their ploidy levels; 2C and 16C nuclei were collected for downstream experiments. As we used a tissue-specific GFP marker, our nuclei consisted of epidermal cell nuclei after tissue dissection. Sorting was performed with Summit 5.2 (Beckman Coulter); re-rendering and basic statistic extraction of the data were done with FCS Express v.6 (deNovo); and graphing and statistical analysis were performed with JMP v.13 (SAS).

### 4.2. ATAC-seq

Our ATAC-seq experiment was performed with two biological replicates. For each replicate, 10,000 sorted 2C and 16C nuclei were collected in 1XPBS buffer and centrifuged at 3000× *g* at 4 °C for 5 min. Then, 20 μL Tn5 transposase (Illumina) reactions were prepared according to [69] and nuclei were resuspended in this master mix. MinElute PCR Purification Kit (Qiagen) used to purify DNA. Selected Nextera index oligos (Illumina) were used for library preparation.

### 4.3. ATAC-seq Reads Processing and Peak Calling

Our ATAC-seq data analysis was performed as previously published with minor differences [44]. Bowtie 2 v2.2.4 was used to align ATAC-seq reads to the *Arabidopsis* thaliana reference genome (TAIR10) [70]. ATAC-seq peaks were identified by using MACS 2 with the following parameters: “nomodel--shift −50--extsize 100--keep-dup = 1” [71]. A list of potentially accessible regions was created by merging the MACS2 identified peaks from all samples and number of raw reads mapped to each region for each condition was quantified using the multicov function in BedTools [72]. This count table was used to identify differentially accessible regions with the DESeq2 package in R [73]. The criteria to achieve differential peaks were log2 fold change greater than 1.5 to call gain-of-accessibility (UP) and loss-of-accessibility (DOWN) peaks and padj smaller than 0.01. For GO analysis, differentially accessible genes from Deseq2 were uploaded to the plant GSEA with default parameters (Appendix A) [74]. The ATAC-seq peaks called from each treatment and replicate, the peak count table, and the differentially accessible peaks can be found in Appendix A. For the motif shown in Figure 3B, 160 bp regions from both sides of the midpoint of the differentially accessible regions are submitted to the MEME tool with the following parameters: “meme sequences.fa-dna-oc. -nostatus-time 18,000-mod zoops-nmotifs 4-minw 6-maxw 15-objfun classic-revcomp-markov_order 0” [49].

### 4.4. Analysis of Gene Regulation in Microarray Datasets

Affymetrix 25k files from AtGenExpress: “Light treatments” TAIR-ME00345 (www.arabidopsis.org) were imported into GeneSpring GX and GC-RMA normalized using the onboard “GC RMA File Processor” algorithm. Thereafter, the total dataset was normalized “specific samples per gene” to the dark control. Genes were inspected for two-fold changes over the dark control due to white light samples cross-comparing all time points (Appendix A). TAIR9 was used as the genome reference.

## Figures and Tables

**Figure 1 plants-09-01478-f001:**
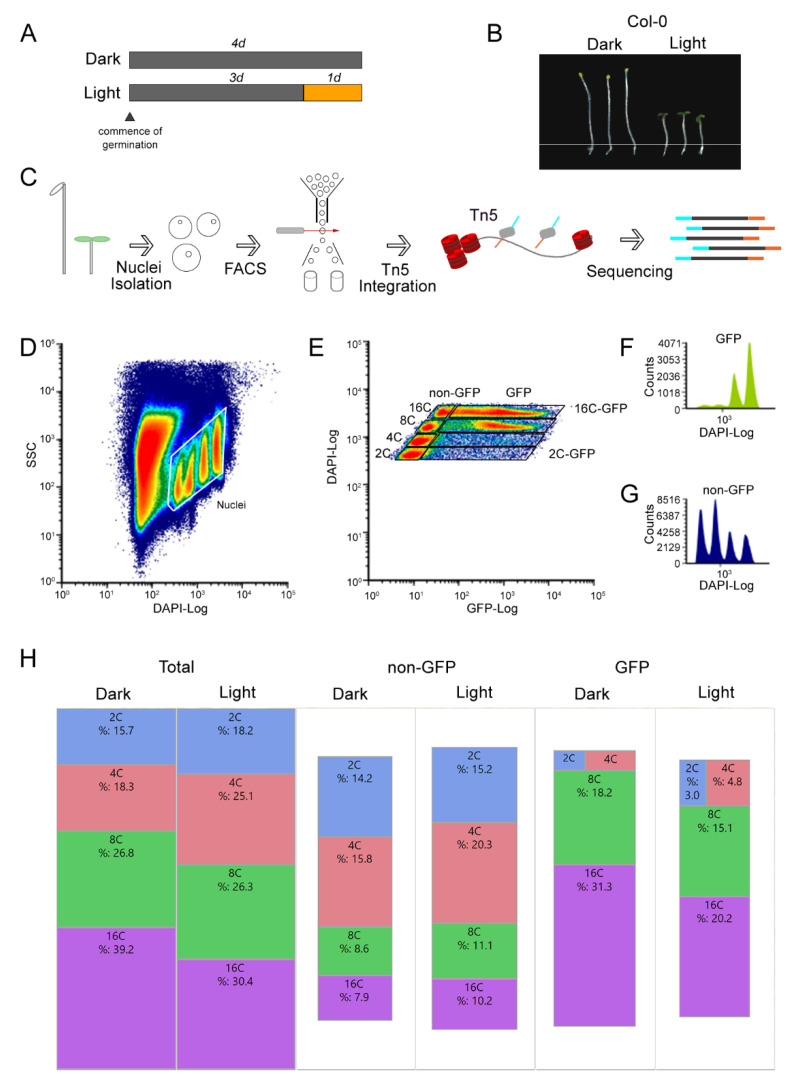
Endopolyploidy levels increase in dark-grown hypocotyls specifically in the epidermal layer. (**A**) Experimental design for light and dark-grown hypocotyls. (**B**) Representative images of light- and dark-grown Col-0 hypocotyls. (**C**) Workflow of fluorescence activated nuclei sorting (FANS) combined with the assay for transposase-accessible chromatin using sequencing (ATAC-seq). (**D**) Bivariate density histogram of the initial gate used to separate nuclei from major debris. See methods for further channel clean up. (**E**) Bivariate density histogram of the final gates used for sorting after clean-up for all endoploidy regions. (**F**) DAPI signal for all nuclei expressing the GFP marker. (**G**) DAPI signal for all nuclei expressing the not expressing the GFP marker. (**H**) TreeMap view of the percentage of nuclei for each endoreduplication level for a typical sort. The size of the rectangles is proportional to the sample size.

**Figure 2 plants-09-01478-f002:**
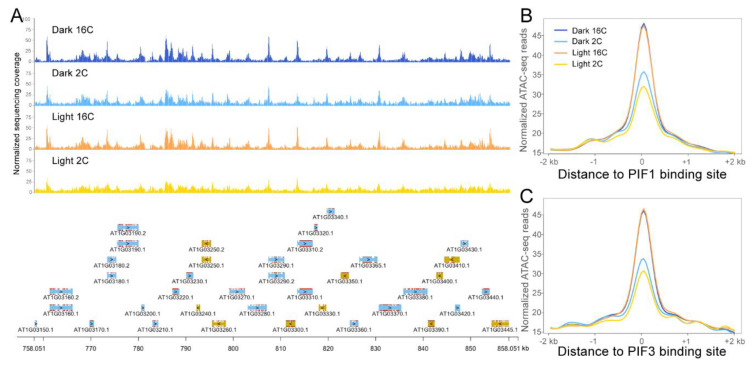
Chromatin accessibility patterns of 2C and 16C nuclei from dark and light treated hypocotyls. (**A**) Representative ATAC-seq peaks of different samples from a 100 kb genomic region of Chr1. In the panel below, genes indicated by blue are on the “Watson” strand and yellow ones are on the “Crick” strand. ATAC-seq read density is plotted with 50 bp windows. Normalized ATAC-seq signals around PIF1 (**B**) and PIF3 (**C**) transcription factor (TF) binding sites.

**Figure 3 plants-09-01478-f003:**
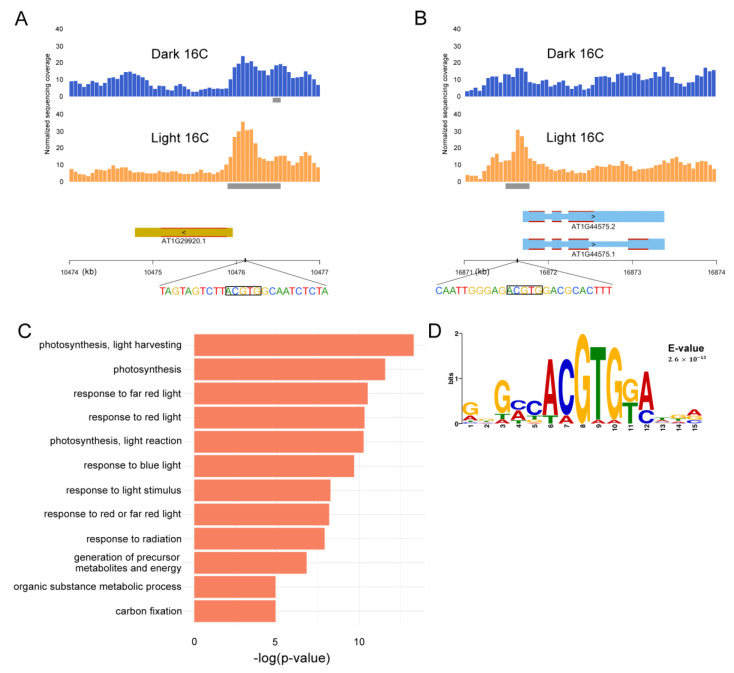
Analysis of differentially accessible genes and regions of dark and light hypocotyls. (**A**) Differential light accessible regions (LAR) ATAC-seq peaks around the promoter of *CHLOROPHYLL A/B-BINDING PROTEIN 2* (*CAB2*) and (**B**) *NONPHOTOCHEMICAL QUENCHING 4* (*NPQ4*). Both genes are on Chr1. Grey bars indicate ATAC-seq peaks annotated by MACS2. (**C**) Gene Ontology (GO) analysis of the 56 genes that are more accessible in light treated hypocotyl in 16C nuclei. (**D**) Sequence Logo of the recovered G-Box motif from the Multiple Em for Motif Elicitation (MEME) motif analysis of all LARs.

**Figure 4 plants-09-01478-f004:**
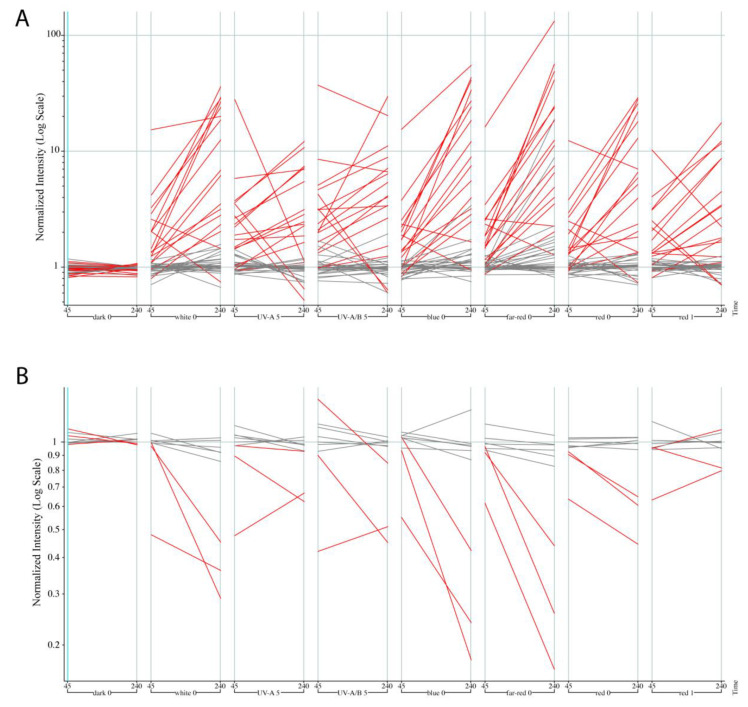
Regulation by light for LAR and DAR genes in the AtGenExpress: “Light Treatments” experiment. (**A**) The 46 out of 56 LAR genes found in the dataset, with seventeen two-fold up-regulated in white light, highlighted in red. (**B**) The 9 out of 11 DAR genes found in the dataset, with three two-fold down-regulated in white light, highlighted in red.

## Data Availability

Short read data are publicly available at the NCBI Sequence Read Archive as BioProject accession number PRJNA664913.

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
