# Peer review of "Isolation of Lineage Specific Nuclei Based on Distinct Endoreduplication Levels and Tissue-Specific Markers to Study Chromatin Accessibility Landscapes"

_plants, 2020, doi:10.3390/plants9111478_

Round 1
Reviewer 1 Report
Manuscript ID: plants-967936
Type of manuscript: Article
Title: Isolation of Lineage Specific Nuclei Based on Distinct Endoreduplication Levels and Tissue-specific Markers to Study Chromatin Accessibility Landscapes
Authors: Ezgi Süheyla Karaaslan, Natalie Faiss, Chang Liu, Kenneth Wayne Berendzen
Dear Authors,
The research is coherent to the scope of the Special Issue "Chromatin Dynamics for Developmental Transitions in Plants". The Abstract is clearly described and comprehensive. The Introduction is also straightforward and clear. The aim of the work was not clearly defined and no research hypothesis was formulated. Although the authors clearly presented the results, they did not come to the proper conclusions.
Details about in vitro experimental design are missing. Please add information about disinfection, how many seeds were used for experiment.
Section Results and Discussion should be rewritten. When describing the results of their research the authors refer to the literature and the discussion is largely a repetition of the results. We can find there citation of the tables and figures talk over in the results. It makes it difficult for the reader to follow the results and is confusing. Please remove sentences that do not represent the authors' results from the results and include them in the discussion. See the manuscript.

Reviewer 2 Report
In this manuscript, the authors observe differential chromatin changes in the 2C and 16C of epidermal cells and, interestingly, find very few of them. They further identify the genes having light-dependent chromatin accessibility changes in the 16C nuclei that are paralleled by light-dependent transcriptional changes. Thus, in their results, transcriptional reprogramming can directly be associated with changes in the chromatin accessibility, at least in the 16C cells.
I believe the MS is rich in original data obtained using an interesting approach. However, the MS should be partially rewritten to clarify its main message (see below)
Major comment:
The current version of Introduction does not show any relatedness with the Abstract (why the isolation of lineage-specific nuclei of different ploidies is needed for this particular study). The authors should consider what is their priority in this manuscript: to describe the optimized method of FANS to analyse possible differences in chromatin structure (as it appears from the Title and partially from the Abstract), or to study light-dependent regulation of gene expression by transcription factors (as the Introduction suggests, without mentioning the FANS approach at all). Briefly, please, make these three items (Title, Abstract, Introduction) mutually more compatible and clarify what is the main message of the MS. While doing this, please also add a paragraph to the end of the Introduction where the aim and approach of the study would be presented (In this study, we...).
Minor comments:
- Abbreviations should be explained on their first appearance (e.g., ATAC-seq...)
- Line 192: “publically“ instead of “publicly“
- Line 232: “This“ instead of “Thus“
Round 2
Reviewer 2 Report
In the revised version, the authors have improved the quality of presentation of the research background and the results. Also Discussion is better elaborated in the present form.
Author Response
In the revised version, the authors have improved the quality of presentation of the research background and the results. Also Discussion is better elaborated in the present form.
-Thank you.